# Changes in Healthcare Utilization in Japan in the Aftermath of the COVID-19 Pandemic: A Time Series Analysis of Japanese National Data Through November 2023

**DOI:** 10.3390/healthcare12222307

**Published:** 2024-11-19

**Authors:** Yuta Tanoue, Alton Cao, Masahide Koda, Nahoko Harada, Cyrus Ghaznavi, Shuhei Nomura

**Affiliations:** 1Faculty of Marine Technology, Tokyo University of Marine Science and Technology, Tokyo 135-8533, Japan; ytan001@kaiyodai.ac.jp; 2Department of Global Health Policy, Graduate School of Medicine, The University of Tokyo, Tokyo 113-0033, Japan; altoncao@gmail.com; 3Department of Health Policy and Management, Graduate School of Medicine, Keio University School of Medicine, Tokyo 160-8582, Japan; cghaznavi@gmail.com; 4Co-Learning Community Healthcare Re-Innovation Office, Graduate School of Medicine, Dentistry and Pharmaceutical Sciences, Okayama University, Okayama 700-8558, Japan; shoei05@gmail.com; 5Department of Nursing Science, Graduate School of Interdisciplinary Science and Engineering in Health Systems, Okayama University, Okayama 700-0914, Japan; 6Department of Medicine, University of California San Francisco, San Francisco, CA 94143, USA; 7Keio University Global Research Institute (KGRI), Tokyo 108-8345, Japan

**Keywords:** COVID-19, healthcare access, healthcare capacity, hospital beds, hospital stay, outpatient visit, Japan

## Abstract

Introduction: The COVID-19 pandemic precipitated substantial disruptions in healthcare utilization globally. In Japan, reduced healthcare utilization during the pandemic’s early phases had been documented previously. However, few studies have investigated the impact of the pandemic’s later stages (2022–2023) on healthcare utilization rates, particularly in the Japanese context. Methods: We employed a quasi-Poisson regression model, adapted from the FluMOMO framework, to analyze temporal trends in Japanese healthcare utilization throughout the pandemic until November 2023. We estimated inpatient and outpatient volumes and hospital length of stay by bed type (general and psychiatric). Results: In general hospital beds, inpatient volumes remained significantly below pre-pandemic levels for every month until November 2023, with a reduction of 7.8 percent in 2023 compared to pre-pandemic levels. Psychiatric inpatient volumes, which had been declining before the pandemic, continued this downward trend, with the average occupancy rate decreasing by approximately 5.3% to 81.3% in 2023 compared to pre-pandemic levels. Significantly reduced outpatient volumes for both general and psychiatric care, in addition to prolonged lengths of hospital stay for psychiatric beds, were observed sporadically for several months in 2022 and 2023, persisting beyond the cessation of state of emergency and quasi-state of emergency declarations. Conclusion: The COVID-19 pandemic fundamentally altered healthcare utilization patterns in Japan. We observed a sustained reduction in general and psychiatric inpatient volumes relative to pre-pandemic baselines nationwide. The prolonged impact on healthcare utilization patterns, persisting beyond emergency measures, warrants continued monitoring of service delivery.

## 1. Introduction

### 1.1. Background

Despite the World Health Organization (WHO) declaring the end of the COVID-19 public health emergency on 5 May 2023, the pandemic’s widespread impacts and longitudinal collateral effects will continue to be felt for years. The pandemic shocked health systems globally through direct burdens on limited resources and indirect disruptions to healthcare delivery [1]. Foregone care secondary to the pandemic has been widely documented, with factors such as a lack of medical resources, fear of nosocomial infections, and stigma contributing to declining healthcare utilization for non-COVID-19 individuals [2,3,4,5,6,7]. Concurrently, telemedicine has proliferated globally [6,8,9,10,11,12].

Japan, known for its accessible healthcare system, experienced similar disruptions. Previous studies with national data up to October 2021 showed significantly attenuated outpatient healthcare service utilization early in the pandemic, particularly during state of emergency declarations, before returning to pre-pandemic levels in late 2021 [2,13]. Since January 2020, Japan has declared four states of emergency and two “Manbou periods” (“quasi-states of emergency”), with the last concluding in February 2022. Inpatient volumes decreased significantly and did not return to pre-pandemic levels throughout the study period.

### 1.2. Related Studies

Foregone care and essential service disruption were widely studied during the pandemic. Within the global context, it was reported by the WHO that 84% of 124 surveyed countries were still experiencing some reduction in service volume for at least one essential medical service at the end of 2022; this represented significant efforts made toward recovery, with 23% of services disrupted on average per country in late 2022 compared to 56% in 2020 [14]. In the most recent global systematic review on the impact of the pandemic on healthcare utilization, using data up until August 2020, a median 37% reduction in overall medical services, a 42% reduction in outpatient visits, and a 28% reduction in inpatient admissions were identified [15]. However, there is a significant gap in the literature, with few studies published on healthcare utilization recovery since late 2021 following the end of the most acute phases of the pandemic with global lockdowns, and even fewer past the declared end to the pandemic in May 2023. In one study from South Korea, a regional peer to Japan with similarly low COVID-19-associated mortality, most health services found that their utilization levels rebounded to pre-pandemic levels by late 2022 [16]. On the contrary, a study from the United States (US) in a large health system in California showed the opposite pattern, with maintained exiguous utilization for inpatient and emergency services after the end of the pandemic in June 2023 [6].

### 1.3. Objectives

Different from many other high-income countries, Japan experienced its heaviest COVID-19 burden after 2021, with the rapid spread of the Omicron variant and peaks of COVID-related hospitalizations between January 2022–March 2022, July 2022–September 2022, and November 2022–January 2023, coinciding with significant increases in excess mortality [17,18]. However, the impact of these surges on healthcare access and utilization from late 2021 until the pandemic’s end in 2023 remains unclear. To quantify long-term changes to healthcare utilization in Japan, before and after the pandemic, our study used Japanese national data from the Ministry of Health, Labour, and Welfare (MHLW) to evaluate the pandemic’s impact on healthcare utilization in Japan until November 2023. We tracked monthly data on inpatient volume, occupancy rate, hospital beds, new hospitalizations, length of stays, and outpatient volume. This long-term modeling through the pandemic’s lifecycle can provide valuable insights for future policymakers to optimize healthcare system resilience and plan for equitable, efficient recovery.

This paper is structured to first introduce the methods and statistical approach, followed by a presentation of the results and concluding with a discussion that includes sections on policy implications and methodological limitations.

## 2. Materials and Methods

### 2.1. Methodological Workflow

This study analyzed the indirect effects of the COVID-19 pandemic on healthcare utilization in Japan through the following process. First, we obtained monthly hospital report data from the MHLW for general hospital and psychiatric beds from January 2012 to November 2023. The data included key healthcare delivery indicators such as inpatient volume, occupancy rate, number of hospital beds, new hospitalizations, length of hospital stays, and outpatient volume. After careful consideration of the study objectives, we excluded infectious disease beds, tuberculosis beds, and long-term care beds from the analysis to focus on the pandemic’s indirect effects on acute and psychiatric care utilization. Using quasi-Poisson regression modeling based on the FluMOMO approach, we then analyzed pre-pandemic data (2012–2019) to establish baseline utilization patterns and predict expected utilization levels. Finally, we compared these predictions with actual observed values during the pandemic period to quantify healthcare utilization deficits. 

### 2.2. Data Sources

Like past studies from our group [2,13], monthly hospital report data were obtained from the MHLW for this study. Public hospital report data have been available since 1949, documenting basic indicators of healthcare delivery and utilization, stratified by prefecture. For this study, nationally aggregated, monthly hospital report data for general hospital beds (which included all clinical specialties) and psychiatric beds were utilized from January 2012 to November 2023 (past the end of the WHO-defined pandemic period) for the following variables: inpatient volume, occupancy rate, number of hospital beds, new hospitalizations, length of hospital stays, and outpatient volume. Of note is that neither the statistics provided for inpatient nor outpatient volume included telehealth visits. Statistical analyses were then stratified according to these bed types, and modeled monthly counts were temporally compared with COVID-19 cases and emergency declarations. 

Previous studies also included three other bed types: tuberculosis beds, long-term care (LTC) beds covered by LTC insurance, and all LTC beds. To best focus this study on the indirect effects of the pandemic on healthcare utilization, tuberculosis beds were excluded from the analysis as they were typically the first bed type, aside from dedicated infectious disease beds, to service COVID-19 patients. The utilization patterns for LTC beds were also excluded due to the usage patterns being dominated by the influence of national policies to shift end-of-life care and management to community care [19]. Furthermore, there are concerns regarding the misclassification of some community care beds as LTC beds due to the absence of unified international definitions for bed types [20]. Therefore, these bed types were excluded from the analysis performed in this study. 

### 2.3. Statistical Methods

To investigate the impact of the pandemic on healthcare utilization, we first developed a model to estimate any significant differences between the expected utilization based on long-term trends with the actual recorded values. We extrapolated expected monthly counts for inpatient and outpatient volumes and the monthly average duration of hospital stays, represented as *µ_i_* in the following formula, with the following quasi-Poisson regression model based on the FluMOMO model first described in Nielsen et al. 2018 [21]. This method, which has been widely adopted in various epidemiological studies including analyses of excess mortality in Europe [22,23], offers a more straightforward and elegant alternative to the Farrington algorithm used in previous studies. The Farrington model, while commonly used in epidemiological studies, has limitations for our analysis due to its reliance on multiple analyst-defined parameters, including seasonal adjustment factors. This introduces potential subjectivity, as different analysts may select different parameters. Furthermore, the model offers multiple methods for calculating confidence intervals, which can lead to inconsistent results. In contrast, the FluMOMO method we adopted requires no arbitrary parameter selection by analysts, ensuring more objective and reproducible results [2,13]. It also effectively captures seasonality through sine and cosine functions with a periodicity of one and a half years. The formulae for our model are as follows.
(1)$$\begin{array}{c} \log\left(\mu_{i}\right)=b_{0}+b_{1}t_{i}+b_{\sin12}\sin\left(\left(2\pi /12\right)t_{i}\right)+b_{\sin6}\sin\left(\left(2\pi /6\right)t_{i}\right)+\\ b_{\cos12}\;\cos\left(\left(2\pi /12\right)t_{i}\right)+b_{\cos6}\cos\left(\left(2\pi /6\right)t_{i}\right),\; \end{array}$$
(2)$$\phi =\max\left\{1,\frac{1}{n-p}\sum_{i=1}^{n}\frac{{\left(y_{i}-\mu_{i}\right)}^{2}}{\mu_{i}}\right\}$$

Parameters in the model were estimated with the quasi-likelihood method using pre-pandemic data from January 2012 to December 2019. *y_i_* represents the observed counts at time-point *t_i_*, and *ϕ* is the dispersion parameter of the quasi-Poisson model. The observed values of the length of hospital stays were treated as continuous variables and were rounded when the quasi-Poisson model was estimated. *p* represents the degrees of freedom and *n* the number of data points. 

We then estimated the upper bounds *U_i_* and lower bounds *L_i_* of predicted counts for time points *t_i_* with the following formulae. Here, *V*_log_(*µ_i_*) represents the log-scale variance of *µ_i_*.
(3)$$U_{i}={\left(\mu_{i}^{\frac{2}{3}}+1.96\sqrt{V}\right)}^{3/2}$$
(4)$$L_{i}=\max\left\{{\left(\mu_{i}^{\frac{2}{3}}-1.96\sqrt{V}\right)}^{3/2},0\right\}$$
(5)$$V={\left(\frac{2}{3}\mu_{i}^{\frac{2}{3}-1}\right)}^{2}\phi \mu_{i}+{\left(\frac{2}{3}\mu_{i}^{\frac{2}{3}-1}\right)}^{2}V_{\mu_{i}}$$
(6)$$V_{\mu_{i}}=\left(exp\left(V_{log}\left(\mu_{i}\right)\right)-1\right)exp\left(2log\left(\mu_{i}\right)+V_{log}\left(\mu_{i}\right)\right)$$

Finally, percent deficits [a,b], comparing the difference between expected and observed counts for particular observations at 95% intervals of confidence, were also estimated and presented as follows: (7)$$a=\frac{max\left(L_{i}-y_{i},0\right)}{\mu_{i}}\times 100,\;\;b=\frac{\max\left(\mu_{i}-y_{i},0\right)}{\mu_{i}}\times 100$$

## 3. Results

### 3.1. General Hospital Beds

For general hospital beds, the inpatient volume decreased significantly since the beginning of the pandemic in March 2020 (nadiring in May 2020, percent deficit = [11.14–13.75]) and has remained statistically deficient for every month until November 2023, as shown in Figure 1A. On average, there were 626,450 inpatients in general hospital beds in 2023 compared to an average of 679,092 between the years of 2017 and 2019, representing a 7.8% reduction from pre-pandemic levels. Two state of emergency periods, from April 2020 to June 2020 and from July 2021 to September 2021, demonstrated the most prominent monthly deficits in the numbers of inpatients, with an average of 593,547 and 611,727 inpatients, respectively. Inpatient volumes have also remained persistently deficient even without any state of emergency declarations. For example, in August 2022, when the number of COVID-19 infections was at its historic peak during a period without a state of emergency declaration, the number of inpatients maintained a significant deficit (percent deficit = [10.24–12.89]).

With regards to the number of general hospital beds, there was a decreasing trend which tapered off in the latter half of 2021 (Figure 1B). However, this observed decrease in the absolute number of hospital beds remained small, at less than 1% of the total number of beds. 

The number of new hospitalizations, like the inpatient volume, tended to decrease under the state of emergency and quasi-state of emergency declarations (Figure 1C). The yearly average number of new hospitalizations in the first three years of the pandemic from 2020 to 2022 was 1,170,022, while the average annual number of new hospitalizations in the three years before the pandemic from 2017 to 2019 was 1,291,847, resulting in a decrease of about 10% on average between these periods. 

Changes to the bed occupancy rate, i.e., the inpatient utilization rate, were also observed over the pandemic (Figure 1D). The average usage rate for the first three years of the pandemic from 2020 to 2022 was 67.5%, while the average usage rate for the three years before the pandemic from 2017 to 2019 was 73.5%, amounting to a decrease of about 6% between these periods. 

### 3.2. Psychiatric Beds

Inpatient volume in psychiatric beds was found to have been on a downward trend since before the pandemic, which is apparent in the decreasing long-term trend of expected volumes (Figure 2A). However, this downward trend intensified throughout the pandemic. Statistically significant deficient inpatient volumes were first recorded in April and May 2020, which coincided with the first state of emergency declaration. From January 2021 (the start of the second state of emergency declaration), nearly all months until the end of the study period in November 2023, well past the declared end to the public health emergency, demonstrated statistically significant deficiencies in monthly inpatient volumes. 

The total number of psychiatric beds decreased more substantially compared to general hospital beds, at a nearly constant rate from before the pandemic until the end of the study period in November 2023, as shown in panel B of Figure 2. There was a reduction of over 3% in the number of psychiatric beds from an average of 330,089 beds between 2017 and 2019 before the pandemic to 320,170 beds in 2023. There was no significant observed difference in the pace of decline before and after the onset of the pandemic.

The number of new hospitalizations in psychiatric beds was also found to decline with declarations of a state of emergency and a quasi-state of emergency, similar to the trend for general hospital beds (Figure 2C). The yearly average number of new hospitalizations in the first three years of the pandemic from 2020 to 2022 was 29,475, compared to an average of 32,162 before the pandemic from 2017 to 2019, amounting to a decrease of about 8% between these periods.

Regarding the occupancy rate of psychiatric beds, an accelerating downward trend was observed for the duration of the pandemic period (Figure 2D). The average occupancy rate fell by about 5.3%, from 85.6% before the pandemic between the years of 2017 and 2019 to 81.3% in 2023. 

Overall, unlike general hospital beds, there was a notable long-term decline in both the number of psychiatric beds and the occupancy rate. The combined effect of these factors resulted in the significant under-occupation of psychiatric beds. Although the decrease in the number of beds was observed to be an approximately linear long-term trend, the magnitude of the loss of beds was more substantial than that of general hospital beds, and the decrease in inpatient utilization in psychiatric beds was significantly exacerbated with the onset of the pandemic. 

### 3.3. Outpatient Volumes

Since the beginning of the pandemic, there have been significant deficits in the outpatient volume of general hospital and psychiatric beds during the declarations of the first, second, and third states of emergency (April 2020–May 2020, January 2021–March 2021, and April 2021 to June 2021, respectively), as seen in Figure 3A,B. Additionally, significantly deficient months continued to be sporadically observed in a few months in 2022 and 2023, past the last declarations of states of emergency or quasi-states of emergency. In particular, January 2023 had significantly low outpatient volumes for both general hospital beds (percent deficit = [2.56–9.34]) and psychiatric beds (percent deficit = [2.77–9.60]), coinciding with the large peak of infected COVID-19 patients in late 2022. 

### 3.4. Length of Hospital Stay

For general hospital beds, there was no significant change in the number of days of hospitalization throughout the entire pandemic period (Figure 4A). However, for psychiatric beds, as shown in Figure 4B, excess hospital stay lengths were observed during the first, second, and third state of emergency declarations and the second declaration of a quasi-state of emergency. Furthermore, significant excesses were occasionally observed after the last declaration of a quasi-state of emergency, usually coinciding with peaks of COVID-19 infections (May 2022, July 2022, August 2022, and January 2023). 

The average number of days hospitalized in psychiatric beds in the first three years of the pandemic from 2020 to 2022 was approximately 276 days, 10 days longer than the average of 266 days of hospitalization in the three years before the pandemic from 2017 to 2019. 

## 4. Discussion

This study tracked Japanese healthcare utilization until November 2023, over two years after the last state of emergency and after COVID-19 was downgraded in Japan to the same severity level as seasonal influenza in May 2023. With regard to the methodology of this study, the FluMOMO model was chosen due to the necessity to estimate expected baseline patterns of health utilization. While machine learning methods often have high accuracy in predicting average values, it is often difficult to theoretically calculate the upper and lower limits of predictions. In addition, methods using quasi-Poisson regression, such as the FluMOMO model and the Farrington model, have the advantage of being able to theoretically handle overdispersion appropriately and are widely used in the field of public health.

Our analysis revealed that inpatient volume continues to decline at an accelerated rate compared to pre-pandemic levels for both general and psychiatric beds, with a more prominent decline in psychiatric beds. Modeling showed that inpatient volume declines exceeded the prior trend, recording consistently low volumes for nearly all months since the onset of the pandemic. This acceleration, combined with a reduction in psychiatric beds, likely indicates a sustained decreasing trend for overall inpatient volumes in Japan. 

Early studies on the COVID-19 outbreak in Japan up to November 2020 also observed a significant decline in medical service use, attributed to patients avoiding visits, efforts to secure hospital beds, and revisions to prevent infection spread within hospitals [24]. While outpatient volumes have largely recovered to pre-pandemic levels since the second half of 2021, there continue to be months of low utilization for general and psychiatric beds into 2022 and 2023, likely associated with COVID-19 infection peaks.

Our literature review on healthcare utilization during the late pandemic phase revealed varying recovery patterns across regions. The foundational systematic review by Moynihan et al. documented substantial global disruptions during the early phases of the pandemic, with median reductions of 37% across all services, demonstrating the widespread nature of initial service impacts. Their finding that reductions were generally larger among those with milder conditions suggests a degree of potential protective triaging of more severe cases [15]. More recent WHO data show the first major signs of global service recovery, with the average extent of disruption decreasing from 56% of services in 2020 to 23% by late 2022, though 84% of countries still report some disruption [14]. A comprehensive US study found that while overall visit rates, including telehealth, had recovered by late 2022, inpatient and emergency medicine utilization remained 7.5% and 8% below pre-pandemic levels among non-COVID-19 patients [6]. In South Korea, most health services rebounded to pre-pandemic levels by late 2022 after initial decreases of 15.7% for outpatients and 11.6% for inpatients in early 2020, though recovery varied across services and population groups [16]. Our extended monitoring period through 2023 builds upon these findings, demonstrating that while some service areas showed recovery, others experienced lasting changes in utilization patterns that persisted well beyond the acute phase of the pandemic.

A similar trend was observed in mental healthcare in the US, where inpatient visits recovered to only 79.9% of pre-pandemic levels by August 2022, but overall utilization increased by 38.8%, driven by a remarkable 1068.3% surge in telehealth use [9]. This mirrors global patterns of increased mental health service demand, with studies from multiple countries showing rises in anxiety, depression, and other mental health concerns requiring treatment. The pandemic appears to have particularly impacted patients with pre-existing conditions such as chronic pain, who faced both disrupted access to regular care and increased psychological distress [25]. We posit that Japan likely experienced a comparable scenario, given our findings of persistently reduced in-person patient volumes coupled with the increased adoption of alternative healthcare delivery methods such as telemedicine. This aligns with Japanese MHLW data, which reported a tenfold increase in new patient telemedicine visits from April 2020 to August 2022, alongside a rise in medical institutions offering telemedicine services from 9.7% to 15.8% [26].

Regarding hospital stays, we observed no change in length for general hospital beds, but psychiatric bed stays increased by an average of 10 days compared to pre-pandemic levels. This increase was unevenly distributed, with most excess stays coinciding with emergency declarations and COVID-19 infection peaks until January 2023. The paradoxical trend of decreasing psychiatric bed utilization alongside increasing hospitalization duration during high COVID-19 burden periods likely indicates a disruption of normal discharge planning processes [27].

Notably, Japan experienced its largest COVID-19 burden following the Omicron variant’s spread in 2022 and early 2023, well after the final state of emergency declaration [18]. Despite higher infection rates than in 2020–2021, compliance with preventive measures remained high in Japan [28,29]. This suggests that cautious behaviors, such as delaying psychiatric patient discharges, may have persisted without direct emergency declarations.

Several factors may have contributed to extended psychiatric hospitalizations. Visitation restrictions likely postponed discharge planning conferences, which are crucial for addressing patients’ unique psychosocial needs. Additionally, creating isolation areas in psychiatric wards posed significant challenges [30]. A nationwide survey revealed that about 60% of COVID-19-infected psychiatric patients could not be transferred to other hospitals, partly due to social stigma [31]. This often necessitated isolation within inadequately equipped psychiatric hospitals, potentially leading to poorer outcomes and longer stays for this vulnerable group. Moreover, restrictions on outings and activities may have exacerbated patient symptoms, necessitating further isolation and extended stays [32]. However, the absence of excess hospital stays for psychiatric beds after January 2023 suggests that the pandemic’s impact on hospitalization length and long-term deinstitutionalization efforts may not be permanent.

## 5. Implications

Our findings on sustained changes in healthcare utilization patterns have practical implications for healthcare system planning in Japan. These documented changes in inpatient volumes and psychiatric care utilization can inform evidence-based discussions among policymakers and healthcare administrators about post-pandemic healthcare delivery and resource allocation. The persistent reduction in inpatient volumes and changes in psychiatric care patterns warrant particular attention as Japan’s healthcare system continues to evolve. The reduction in inpatient service utilization in Japan during the late pandemic period could be viewed as a positive step towards healthcare system efficiency under normal circumstances. However, given the substantial excess deaths recorded during this time [33], we cannot rule out potential adverse effects on population health outcomes resulting from this decrease in inpatient volumes.

Despite Japan having one of the highest total hospital bed capacities per capita globally, the most recent data from 2021 indicate that its intensive care bed capacity remains below the OECD average (14.4 beds per 100,000 population compared to the OECD average of 16.9/100,000) [34]. The Japanese healthcare system’s inability to effectively distribute resources during the pandemic has been subject to significant scrutiny. For instance, during the first state of emergency in April–May 2020, the Tokyo metropolis, with its 14 million residents and 1100 ICU beds, had to rely on neighboring prefectures to support the treatment of 105 critically ill COVID-19 patients (0.75 per 100,000 people). This occurred despite a recent analysis estimating an effective capacity of 1.5 severely ill COVID-19 patients per 100,000 people [35]. Several factors have been attributed to this capacity shortfall, including inadequate central institutional powers to coordinate and mobilize medical resources and poor coordination among the small private-sector hospitals that comprise the majority of the country’s hospital beds [35,36,37].

This discrepancy between primary metrics used to assess Japan’s healthcare capacity (such as the overall number of hospital beds) and the system’s actual ability to accommodate patients during a pandemic crisis can lead to misaligned policymaking and recommendations. Further research is necessary to elucidate the causes of excess deaths during the pandemic, including potential downstream impacts of the sustained low levels of inpatient care identified in this study.

As Japan continues its efforts towards deinstitutionalization and transitioning long-term care to community healthcare facilities, policymakers must be mindful of potential adverse health outcomes associated with foregone care and deinstitutionalization, balancing these concerns with economic priorities. Despite the post-pandemic proliferation of telehealth and the reduction of financing barriers for Japanese healthcare providers [38], telehealth access and utilization in Japan remain relatively low compared to other high-income nations [39,40]. Furthermore, a recent study from the US found a slight decline in telehealth availability (81.6% to 79.0%) among mental health providers by late 2023, after the end of pandemic-era telehealth-promoting programs in the US [41]. This exemplifies that access to such services is impermanent and can be dictated by a myriad of policy factors. Given that telehealth has been found to be a suitable method of health service delivery in many settings, including mental health provision [42], investment into the establishment, monitoring, and continued maintenance of telehealth systems will be crucial in bridging healthcare access gaps as deinstitutionalization efforts progress in Japan. This is particularly important as Japan’s population rapidly ages and more cost-effective healthcare delivery methods become necessary to maintain equitable access.

## 6. Limitations

Our study has several limitations. The pseudo-Poisson regression analysis used pre-pandemic data from 2012 to 2019, and potential confounding factors affecting healthcare utilization trends were not incorporated into this model. Our analysis covers only general hospital beds and psychiatric beds, not all hospital bed types, due to the significant effects of recent health policy changes on the reported numbers of other bed types. It is worth noting that during the pandemic, some patients who would typically be admitted to general beds (e.g., older patients with chronic disease exacerbation or acute symptoms) may have been redirected to infectious disease wards due to COVID-19 co-infection or infection control measures, potentially contributing to the observed reduction in general bed utilization. The use of national tabulations as our data source precluded regional subgroup analyses, preventing the examination of geographical heterogeneity in healthcare access. Specific reasons for admission were not available in this data source, so we could not determine trends in the prevalence of specific pathologies or diagnoses, or their associated healthcare access patterns. To our knowledge, statistics on total telehealth utilization in Japan have not been published by the MHLW, leaving this segment of healthcare service engagement unclear. While our examination of healthcare utilization trends has facilitated conjectures regarding the pandemic’s impact, causality cannot be inferred from this analysis and further studies are necessary to explore these relationships.

## 7. Conclusions

The COVID-19 pandemic has fundamentally altered healthcare utilization patterns in Japan, with impacts persisting beyond both the final state of emergency declaration in early 2022 and the WHO’s declared end to the public health emergency in May 2023. Our analysis through November 2023 revealed three key findings: persistent reductions in inpatient volumes for both general hospital and psychiatric beds; significant changes in psychiatric care patterns characterized by prolonged lengths of stay during infection peaks and reduced occupancy rates; and sporadic periods of reduced outpatient volumes coinciding with Omicron variant surges. While reduced inpatient volumes might have been viewed favorably before the pandemic as a sign of improved healthcare system efficiency, the context of documented excess mortality during this period raises important questions about potential adverse health impacts. These sustained alterations, persisting even after the relaxation of emergency measures, suggest fundamental shifts in healthcare delivery patterns that warrant careful monitoring, particularly as Japan continues its broader healthcare system reforms, including deinstitutionalization efforts and the expansion of community-based care. Further research must examine the relationship between reduced inpatient services and population health outcomes, while also investigating the role of alternative care delivery methods, such as telemedicine, in ensuring equitable healthcare access in post-pandemic Japan.

## Figures and Tables

**Figure 1 healthcare-12-02307-f001:**
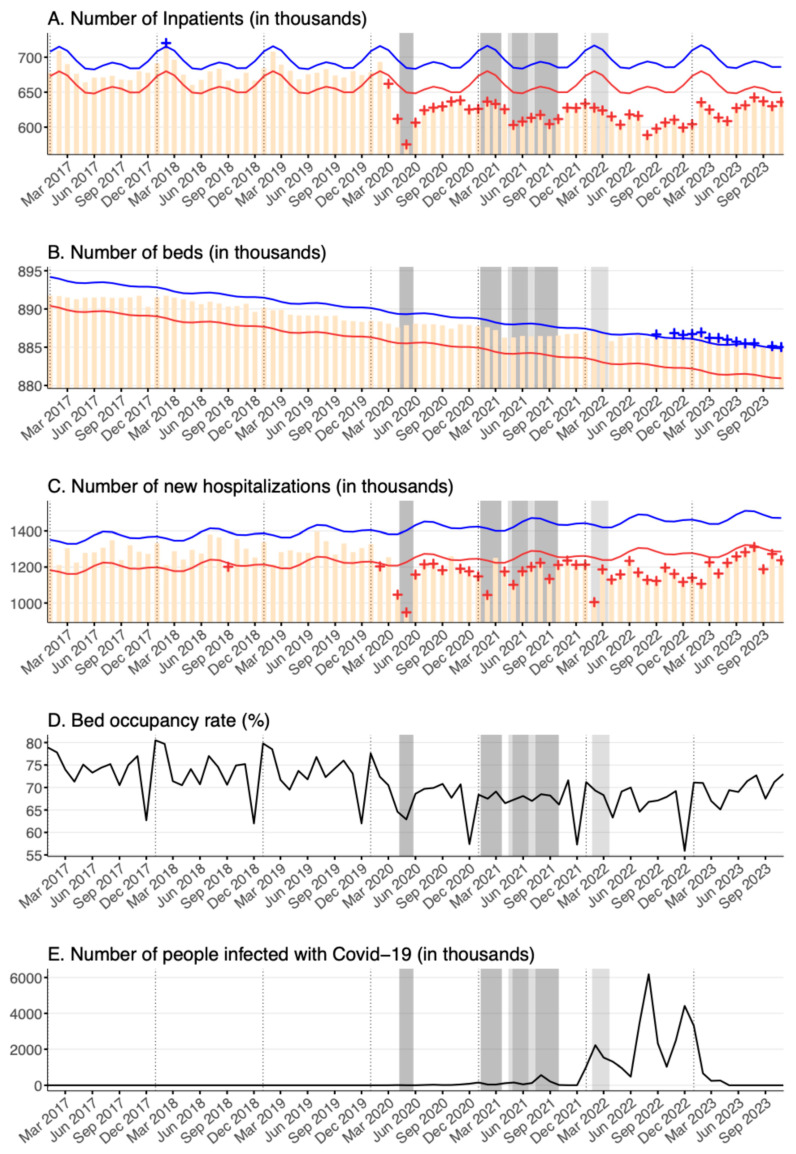
The figure displays the following graphs for general hospital beds between January 2017 and November 2023 (in descending order): (**A**) the number of inpatients (in thousands); (**B**) the number of beds (in thousands); (**C**) the number of new hospitalizations (in thousands); (**D**) the bed occupancy rate (%); and (**E**) the number of people infected with COVID-19 in Japan (in thousands). Yellow bars represent the observed number of inpatients per month. Blue and red lines indicate the upper and lower bounds of the 95% prediction interval, respectively. Statistically significant monthly deficits are marked with red + symbols, and statistically significant monthly excesses are marked with blue + symbols. Black lines in the figures for bed occupancy rate (%) and number of people infected with COVID-19 in Japan indicate the observed rate and number. Bed occupancy rates are “rates”, not counts. The quasi-Poisson regression model based on the FluMOMO model used in this study is a model for count data and cannot be applied to “rate” data. Therefore, no lower or upper limits were calculated for bed occupancy rates. Dark gray shading indicates a declaration of a state of emergency, while light gray shading denotes a declaration of a quasi-state of emergency, both specific to Tokyo. Note that for all panels except the one displaying the number of COVID-19 infections, the y-axes do not start from zero.

**Figure 2 healthcare-12-02307-f002:**
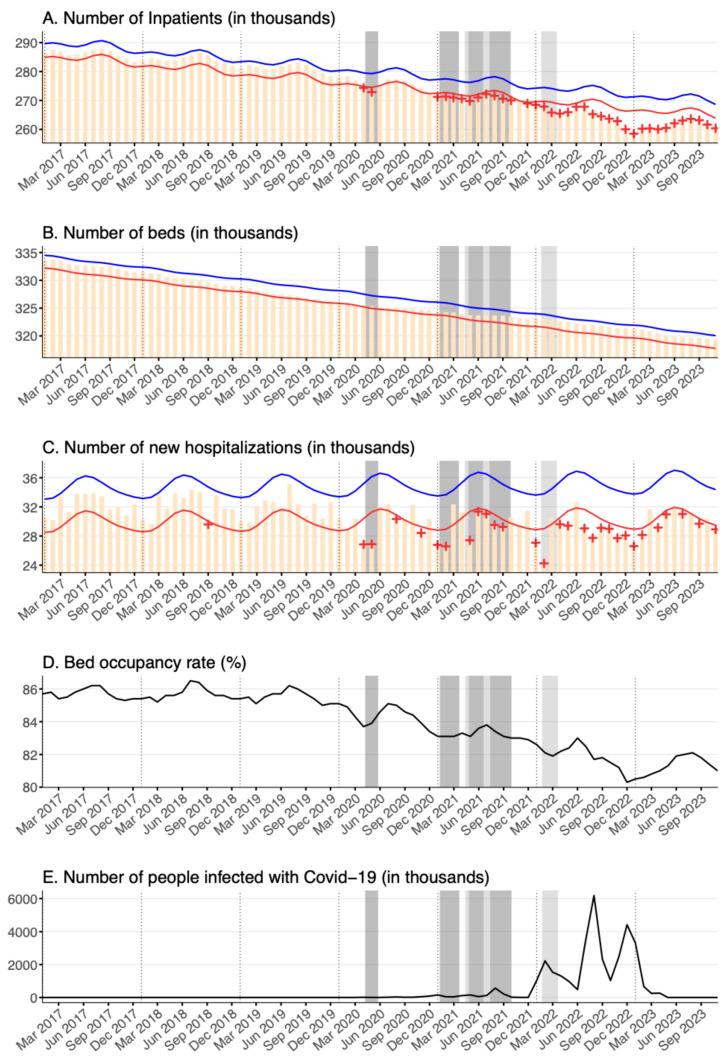
The figure displays the following graphs for psychiatric beds between January 2017 and November 2023 (in descending order): (**A**) the number of inpatients (in thousands); (**B**) the number of beds (in thousands); (**C**) the number of new hospitalizations (in thousands); (**D**) the bed occupancy rate (%); and (**E**) the number of people infected with COVID-19 in Japan (in thousands). Yellow bars represent the observed number of inpatients per month. Blue and red lines indicate the upper and lower bounds of the 95% prediction interval, respectively. Statistically significant monthly deficits are marked with red + symbols, and statistically significant monthly excesses are marked with blue + symbols. Black lines in the figures for bed occupancy rate (%) and number of people infected with COVID-19 in Japan indicate the observed rate and number. Bed occupancy rates are “rates”, not counts. The quasi-Poisson regression model based on the FluMOMO model used in this study is a model for count data and cannot be applied to “rate” data. Dark gray shading indicates the declaration of a state of emergency, while light gray shading denotes the declaration of a quasi-state of emergency, both specific to Tokyo. Note that for all panels except the one displaying the number of COVID-19 infections, the y-axes do not start from zero.

**Figure 3 healthcare-12-02307-f003:**
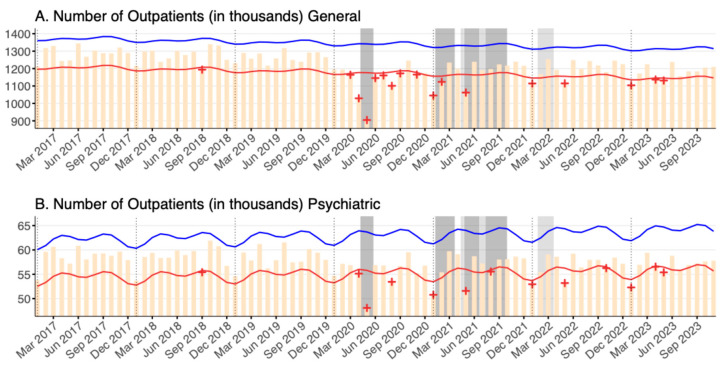
The number of outpatients (in thousands) stratified by bed type in (**A**) general hospital beds and (**B**) psychiatric beds between January 2017 and November 2023. Yellow bars mark the observed number of outpatients per month. Blue and red lines indicate the upper and lower bounds of the 95% prediction interval. Red + symbols indicate statistically significant monthly deficits and blue + symbols are statistically significant monthly excesses. Dark gray shading denotes a declaration of a state of emergency, and light gray shading denotes a declaration of a quasi-state of emergency, both for Tokyo. Note that the y-axes for both panels do not start from zero.

**Figure 4 healthcare-12-02307-f004:**
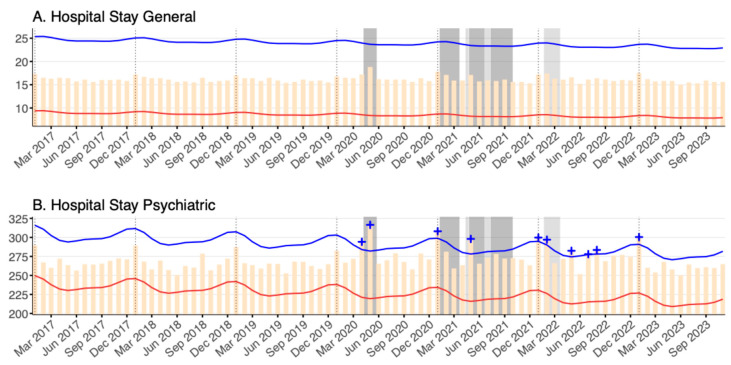
The average length of hospital stays (in days) stratified by bed type in (**A**) general hospital beds and (**B**) psychiatric beds between January 2017 and November 2023. Yellow bars mark the observed average lengths of hospital stays per month. Blue and red lines indicate the upper and lower bounds of the 95% prediction interval. Red + symbols indicate statistically significant monthly deficits and blue + symbols are statistically significant monthly excesses. Dark gray shading denotes a declaration of a state of emergency, and light gray shading denotes a declaration of a quasi-state of emergency, both for Tokyo. Note that the y-axes for both panels do not start from zero.

## Data Availability

The data utilized in this study are publicly available at https://www.mhlw.go.jp/toukei/list/79-1a.html, accessed on 8 October 2024.

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
