# Peer review of "Changes in Healthcare Utilization in Japan in the Aftermath of the COVID-19 Pandemic: A Time Series Analysis of Japanese National Data Through November 2023"

_healthcare, 2024, doi:10.3390/healthcare12222307_

Round 1
Reviewer 1 Report
Comments and Suggestions for Authors
This manuscript is excellent and I think it meets the journal's acceptance criteria.
1. The background and discussion section could have been told in conjunction with more cutting-edge, high-quality literature as a way of highlighting what creative perspectives this study offers.
2. Whose needs are the findings of this study intended to respond to? Please be more explicit and specific about how the findings of this study will be applied in practice.
3. Although there may be no studies focusing on the changes in healthcare resources in Japan after the epidemic, the authors need to put together what is already available in terms of potentially relevant research developments in the background section.
4. The conclusions of this study are derived from speculation and drawing on existing experience. It is recommended that the authors discuss the meaning behind their results in conjunction with more theoretical evidence and more rigorous derivation.
5. The conclusions in the abstract do not just summarise the results, but also briefly analyse the significances and implications of the results.
Author Response
Major comments:
- This manuscript is excellent and| think it meets the journal's acceptance criteria. The only point of suggestion is that the background and discussion section could have been told in conjunction with more cutting-edge, high-quality literature as a way of highlighting what creative perspectives this study offers.
Thank you for your valuable suggestion about incorporating more cutting-edge literature. Following this and another reviewer's suggestion to create a dedicated "Related Studies" subsection in the Introduction, we have restructured the section to better highlight recent recovery patterns across different regions. This reorganization allows us to present the general background of healthcare disruption separately from the focused discussion of recent regional variations, thereby better emphasizing both our study's creative perspectives and the research gap we address regarding post-pandemic healthcare utilization patterns. The modifications were made as follows:
“Related Studies
Foregone care and essential service disruption were widely studied during the pandemic. Within the global context, it was reported by the WHO that 84% of 124 surveyed countries were still experiencing some reduction in service volume for at least one essential medical service at the end of 2022; this represented significant efforts made toward recovery, with 23% of services disrupted on average per country in late 2022 compared to 56% in 2020. [14] In the most recent global systematic review on the impact of the pandemic on healthcare utilization using data up until August 2020, a median 37% reduction in overall medical services, a 42% reduction in outpatient visits, and a 28% reduction in inpatient admissions were identified. [15] However, there is a significant gap in the literature, with few studies published on healthcare utilization recovery since late 2021 following the end of the most acute phases of the pandemic with global lockdowns, and even fewer past the declared end to the pandemic in May 2023. In one study from South Korea, a regional peer to Japan with similarly low COVID-19-associated mortality, most health services found that their utilization levels rebounded to pre-pandemic levels by late 2022. [16] On the contrary, a study from the United States (US) in a large health system in California showed the opposite pattern, with maintained exiguous utilization for inpatient and emergency services after the end of the pandemic in June 2023. [6]” (Introduction, page 2, line 68)
“Our literature review on healthcare utilization during the late pandemic phase revealed varying recovery patterns across regions. The foundational systematic review by Moynihan et al. documented substantial early pandemic disruptions globally, with median reductions of 37% across all services, demonstrating the widespread nature of initial service impacts. Their finding that reductions were generally larger among those with milder conditions suggests some potential protective triaging of more severe cases. [15] More recent WHO data shows the first major signs of global service recovery, with the average extent of disruption decreasing from 56% of services in 2020 to 23% by late 2022, though 84% of countries still report some disruption. [14] A comprehensive US study found that while overall visit rates including telehealth had recovered by late 2022, inpatient and emergency medicine utilization remained 7.5% and 8% below pre-pandemic levels among non-COVID-19 patients. [6] In South Korea, most health services rebounded to pre-pandemic levels by late 2022 after initial decreases of 15.7% for outpatients and 11.6% for inpatients in early 2020, though recovery varied across services and population groups. [16] Our extended monitoring period through 2023 builds upon these findings, demonstrating that while some service areas showed recovery, others experienced lasting changes in utilization patterns that persisted well beyond the acute phase of the pandemic.” (Discussion, page 11, line 351)
“A similar trend was observed in mental healthcare in the US, where inpatient visits recovered to only 79.9% of pre-pandemic levels by August 2022, but overall utilization increased by 38.8%, driven by a remarkable 1068.3% surge in telehealth use. [9] This mirrors global patterns of increased mental health service demand, with studies from multiple countries showing rises in anxiety, depression and other mental health concerns requiring treatment. The pandemic appears to have particularly impacted patients with pre-existing conditions like chronic pain, who faced both disrupted access to regular care and increased psychological distress. [25]” (Discussion, page 11, line 368)
- Whose needs are the findings of this study intended to respond to? Please be more explicit and specific about how the findings of this study will be applied in practice.
Thank you for this important comment. We have revised the implications section to more directly connect our empirical findings to their practical applications, focusing specifically on how the documented changes in healthcare utilization patterns can inform evidence-based healthcare system planning and resource allocation decisions. The modifications were made as follows:
“Our findings on sustained changes in healthcare utilization patterns have practical implications for healthcare system planning in Japan. These documented changes in inpatient volumes and psychiatric care utilization can inform evidence-based discussions among policymakers and healthcare administrators about post-pandemic healthcare delivery and resource allocation. The persistent reduction in inpatient volumes and changes in psychiatric care patterns warrant particular attention as Japan's healthcare system continues to evolve. The reduction in inpatient service utilization in Japan during the late pandemic period could be viewed as a positive step towards healthcare system efficiency under normal circumstances. However, given the substantial excess deaths recorded during this time, [33] we cannot rule out potential adverse effects on population health outcomes resulting from this decrease in inpatient volumes.” (Implications, page 12, line 422)
- Although there may be no studies focusing on the changes in healthcare resources in Japan after the epidemic, the authors need to put together what is already available in terms of potentially relevant research developments in the background section.
Thank you for your valuable comment. As mentioned in our response to Comment 1, we have substantially expanded the Introduction section with additional context and literature review to strengthen the background of our study. For the complete revisions to the Introduction section, please refer to our response to Comment 1.
- The conclusions of this study are derived from speculation and drawing on existing experience. It is recommended that the authors discuss the meaning behind their results in conjunction with more theoretical evidence and more rigorous derivation.
We appreciate your valuable feedback. We have substantially revised the conclusion section to include a more detailed interpretation of our findings, supported by our data analysis and existing literature. Additionally, in response to other reviewers' comments, we have completely restructured the conclusion section to better reflect the empirical evidence and strengthen the analytical framework of our study.
“The COVID-19 pandemic has fundamentally altered healthcare utilization patterns in Japan, with impacts persisting beyond both the final state of emergency declaration in early 2022 and the WHO's declared end to the public health emergency in May 2023. Our analysis through November 2023 revealed three key findings: persistent reductions in inpatient volumes for both general hospital and psychiatric beds; significant changes in psychiatric care patterns characterized by prolonged lengths of stay during infection peaks and reduced occupancy rates; and sporadic periods of reduced outpatient volumes coinciding with Omicron variant surges. While reduced inpatient volumes might have been viewed favorably before the pandemic as a sign of improved healthcare system efficiency, the context of documented excess mortality during this period raises important questions about potential adverse health impacts. These sustained alterations, persisting even after the relaxation of emergency measures, suggest fundamental shifts in healthcare delivery patterns that warrant careful monitoring, particularly as Japan continues its broader healthcare system reforms, including deinstitutionalization efforts and the expansion of community-based care. Further research must examine the relationship between reduced inpatient services and population health outcomes, while also investigating the role of alternative care delivery methods, such as telemedicine, in ensuring equitable healthcare access in post-pandemic Japan.” (Conclusions, page 14, line 500)
- The conclusions in the abstract do not just summarise the results, but also briefly analyse the significances and implications of the results.
Thank you for this valuable suggestion. We have revised the conclusion to not only summarize our key findings but also to highlight their broader implications for healthcare delivery and policy, particularly noting the potential relationship between reduced utilization and population health outcomes that warrants further investigation. The modifications were made as follows:
“Conclusion: The COVID-19 pandemic fundamentally altered healthcare utilization patterns in Ja-pan. We observed a sustained reduction in general and psychiatric inpatient volumes relative to pre-pandemic baselines nationwide. The prolonged impact on healthcare utilization patterns, persisting beyond emergency measures, warrants continued monitoring of service delivery.” (Abstract, page 1, line 36)
Reviewer 2 Report
Comments and Suggestions for Authors
The article is within the aim of the journal and is well focused on deepening health care utilisation long after the pandemic, until November 2023. The authors propose well-discussed data on general and psychiatric health care utilisation; the implications are particularly relevant.
The article is worthy of publication in its present form, although I have minimal issues that could be easily addressed:
Lines 68-70 should probably be removed as stated in the methods
Line 285 is correct 1068% increase in telemedicine use?
A personal reflection on the data, particularly in relation to general hospital under-utilisation. It is studied or possible that patients (old population) typically hospitalised in general beds for chronic disease exacerbation or acute symptomatology, even with infectious co-occurrence according to seasonality, were hospitalised in infectious wards, explaining the reduction in general bed use and occupancy. Alternatively, if infectious beds are within general wards, an under-use could be further sustained, so I suggest to specify in the text what you mean for general wards (general/internal medicine or all clinical specialties)
Many thanks
Author Response
General comments:
- The article is within the aim of the journal and is well focused on deepening health care utilization long after the pandemic, until November 2023. The authors propose well-discussed data on general and psychiatric health care utilisation; the implications are particularly relevant.
Thank you for your positive feedback. Your endorsement of our discussion and implications is highly encouraging.
Major comments:
- Lines 68-70 should probably be removed as stated in the methods
Thank you very much for your comment. We have removed lines 68-70 in accordance with your comment.
- Line 285 is correct 1068% increase in telemedicine use?
Thank you for your comment regarding the cited statistic on line 285. This figure was cited from Cantor 2023 and has been confirmed to be correct.
- A personal reflection on the data, particularly in relation to general hospital under-utilisation. It is studied or possible that patients (old population) typically hospitalised in general beds for chronic disease exacerbation or acute symptomatology, even with infectious co-occurrence according to seasonality, were hospitalised in infectious wards, explaining the reduction in general bed use and occupancy. Alternatively, if infectious beds are within general wards, an under-use could be further sustained, so | suggest to specify in the text what you mean for general wards (general/internal medicine or all clinical specialties)
Thank you for your insightful comment. We have clarified that general hospital beds in our study include all clinical specialties and added a discussion point about potential mechanisms of reduced general bed utilization, specifically noting how older patients with chronic conditions may have been redirected to infectious disease wards due to COVID-19 co-infection or infection control measures during the pandemic. The modifications were made as follows:
“For this study, nationally aggregated, monthly hospital report data for general hospital beds (which included all clinical specialties) and psychiatric beds was utilized from January 2012 to November 2023” (Materials & Methods, page 3, line 127)
“Our study has several limitations. The pseudo-Poisson regression analysis used pre-pandemic data from 2012 to 2019, and potential confounding factors affecting healthcare utilization trends were not incorporated into this model. Our analysis covers only general hospital beds and psychiatric beds, not all hospital bed types, due to the significant effects of recent health policy changes on the reported numbers of other bed types. It is worth noting that during the pandemic, some patients who would typically be admitted to general beds (e.g., older patients with chronic disease exacerbation or acute symptoms) may have been redirected to infectious disease wards due to COVID-19 co-infection or infection control measures, potentially contributing to the observed reduction in general bed utilization.” (Limitations, page 13, line 480)
Reviewer 3 Report
Comments and Suggestions for Authors
The paper presents an interesting and relevant topic. However, it suffers from several drawbacks that need to be corrected before its possible acceptance.
- The introdutor is very very short. The introduction needs more context regarding Covid-19 in Japan and also to frame some of what has been done around the world. If you wish, you can include a literature review chapter where the authors can further develop the theoretical framework, focus on the research problem, taking into account other similar studies
- I advise the authors to professionally proofread their manuscript prior to resubmitting.
- Methods chapter needs more specification of empirical work. The methods are presented in a very technical way, with unnecessary formulas. It would be more important to present the model and the justification for choosing it, to the exclusion of other similar ones. The question of the target population and sample could also be more justified and clarified.
- The results are not badly presented. However, they need a broader discussion with the literature and other similar studies.
- The conclusion chapter is very short.
- I believe there are more studies in indexed journals that use this model or this subjet.
Author Response
General comments:
- The paper presents an interesting and relevant topic. However, it suffers from several drawbacks that need to be corrected before its possible acceptance.
Thank you for your thoughtful feedback. We appreciate your assessment that our topic is interesting and relevant, and we have carefully addressed all your detailed comments to improve the manuscript.
Major comments:
- The introdutor is very very short. The introduction needs more context regarding Covid-19 in Japan and also to frame some of what has been done around the world. If you wish, you can includea literature review chapter where the authors can further develop the theoretical framework, focus on the research problem, taking into account other similar studies
Thank you for the kind suggestion to improve the introduction. We have added a subsection for Related Studies to better contextualize the greater global issues surrounding health utilization and frame our research question. The following text is copied below.
“Related Studies
Foregone care and essential service disruption were widely studied during the pandemic. Within the global context, it was reported by the WHO that 84% of 124 surveyed countries were still experiencing some reduction in service volume for at least one essential medical service at the end of 2022; this represented significant efforts made toward recovery, with 23% of services disrupted on average per country in late 2022 compared to 56% in 2020. [14] In the most recent global systematic review on the impact of the pandemic on healthcare utilization using data up until August 2020, a median 37% reduction in overall medical services, a 42% reduction in outpatient visits, and a 28% reduction in inpatient admissions were identified. [15] However, there is a significant gap in the literature, with few studies published on healthcare utilization recovery since late 2021 following the end of the most acute phases of the pandemic with global lockdowns, and even fewer past the declared end to the pandemic in May 2023. In one study from South Korea, a regional peer to Japan with similarly low COVID-19-associated mortality, most health services found that their utilization levels rebounded to pre-pandemic levels by late 2022. [16] On the contrary, a study from the United States (US) in a large health system in California showed the opposite pattern, with maintained exiguous utilization for inpatient and emergency services after the end of the pandemic in June 2023. [6]” (Introduction, page 2, line 68)
- I advise the authors to professionally proofread their manuscript prior to resubmitting.
We sincerely apologize for the spelling and grammatical errors in our manuscript. Following your valuable feedback, we have had our native English-speaking co-author thoroughly proofread the entire text, including the abstract and Discussion section. All necessary corrections have been made throughout the paper.
- Methods chapter needs more specification of empirical work. The methods are presented in a very technical way, with unnecessary formulas. It would be more important to present the model and the justification for choosing it, to the exclusion of other similar ones. The question of the target population and sample could also be more justified and clarified.
Thank you for your valuable feedback. We have restructured the Methods section by adding the "Methodological Workflow" paragraph to the beginning of Materials & Methods section as suggested by other reviewers. We have also enhanced the justification for our target population selection by explaining why we focused on general hospital and psychiatric beds. Regarding the formulas, while they may appear technical, we believe they are essential for ensuring analytical reproducibility. Additionally, we have added explanation about model selection, noting that the FluMOMO method offers advantages over the Farrington model by eliminating analyst-dependent parameters and providing more objective results.
“Methodological Workflow
This study analyzed the indirect effects of the COVID-19 pandemic on healthcare utilization in Japan through the following process: First, we obtained monthly hospital report data from MHLW for general hospital and psychiatric beds from January 2012 to November 2023. The data included key healthcare delivery indicators such as inpatient volume, occupancy rate, number of hospital beds, new hospitalizations, length of hospital stays, and outpatient volume. After careful consideration of the study objectives, we excluded infectious disease beds, tuberculosis beds, and long-term care beds from the analysis to focus on the pandemic's indirect effects on acute and psychiatric care utilization. Using quasi-Poisson regression modeling based on the FluMOMO approach, we then analyzed pre-pandemic data (2012-2019) to establish baseline utilization patterns and predict expected utilization levels. Finally, we compared these predictions with actual observed values during the pandemic period to quantify healthcare utilization deficits.” (Materials & Methods, page 3 line 109)
“The Farrington model, while commonly used in epidemiological studies, has limitations for our analysis due to its reliance on multiple analyst-defined parameters including seasonal adjustment factors. This introduces potential subjectivity, as different analysts may select different parameters. Furthermore, the model offers multiple methods for calculating confidence intervals, which can lead to inconsistent results. In contrast, the FluMOMO method we adopted requires no arbitrary parameter selection by analysts, ensuring more objective and reproducible results. [2, 13] It also effectively captures seasonality through sine and cosine functions with a periodicity of one and a half years.” (Materials & Methods, page 3, line 154)
- The results are not badly presented. However, they need a broader discussion with the literature and other similar studies.
We appreciate your feedback and have expanded the discussion section to include additional literature that demonstrate similar healthcare utilization patterns and transformations in healthcare delivery models post-pandemic, providing a more comprehensive comparative analysis of our findings. The modifications were made as follows:
“Our literature review on healthcare utilization during the late pandemic phase revealed varying recovery patterns across regions. The foundational systematic review by Moynihan et al. documented substantial early pandemic disruptions globally, with median reductions of 37% across all services, demonstrating the widespread nature of initial service impacts. Their finding that reductions were generally larger among those with milder conditions suggests some potential protective triaging of more severe cases. [15] More recent WHO data shows the first major signs of global service recovery, with the average extent of disruption decreasing from 56% of services in 2020 to 23% by late 2022, though 84% of countries still report some disruption. [14] A comprehensive US study found that while overall visit rates including telehealth had recovered by late 2022, inpatient and emergency medicine utilization remained 7.5% and 8% below pre-pandemic levels among non-COVID-19 patients. [6] In South Korea, most health services rebounded to pre-pandemic levels by late 2022 after initial decreases of 15.7% for outpatients and 11.6% for inpatients in early 2020, though recovery varied across services and population groups. [16] Our extended monitoring period through 2023 builds upon these findings, demonstrating that while some service areas showed recovery, others experienced lasting changes in utilization patterns that persisted well beyond the acute phase of the pandemic.” (Discussion, page 11, line 351)
“A similar trend was observed in mental healthcare in the US, where inpatient visits recovered to only 79.9% of pre-pandemic levels by August 2022, but overall utilization increased by 38.8%, driven by a remarkable 1068.3% surge in telehealth use. [9] This mirrors global patterns of increased mental health service demand, with studies from multiple countries showing rises in anxiety, depression, and other mental health concerns requiring treatment. The pandemic appears to have particularly impacted patients with pre-existing conditions like chronic pain, who faced both disrupted access to regular care and increased psychological distress. [25]” (Discussion, page 11, line 368)
- The conclusion chapter is very short.
Thank you for the reviewer's comment about the conclusion length. We have substantially expanded the conclusion to offer a more comprehensive analysis of our findings. The revised version now provides deeper context about the temporal scope of our study through November 2023, elaborates on the three key findings, discusses their implications within Japan's broader healthcare reform context, and identifies specific areas for future research, including the role of alternative care delivery methods. We believe these additions significantly strengthen the paper's concluding messages while maintaining analytical rigor. The modifications were made as follows:
“The COVID-19 pandemic has fundamentally altered healthcare utilization patterns in Japan, with impacts persisting beyond both the final state of emergency declaration in early 2022 and the WHO's declared end to the public health emergency in May 2023. Our analysis through November 2023 revealed three key findings: persistent reductions in inpatient volumes for both general hospital and psychiatric beds; significant changes in psychiatric care patterns characterized by prolonged lengths of stay during infection peaks and reduced occupancy rates; and sporadic periods of reduced outpatient volumes coinciding with Omicron variant surges. While reduced inpatient volumes might have been viewed favorably before the pandemic as a sign of improved healthcare system efficiency, the context of documented excess mortality during this period raises important questions about potential adverse health impacts. These sustained alterations, persisting even after the relaxation of emergency measures, suggest fundamental shifts in healthcare delivery patterns that warrant careful monitoring, particularly as Japan continues its broader healthcare system reforms, including deinstitutionalization efforts and the expansion of community-based care. Further research must examine the relationship between reduced inpatient services and population health outcomes, while also investigating the role of alternative care delivery methods, such as telemedicine, in ensuring equitable healthcare access in post-pandemic Japan.” (Conclusions, page 14, line 500)
- I believe there are more studies in indexed journals that use this model or this subjet.
Response to reviewer: Thank you for your suggestion. We have strengthened our methodology section by adding references to two additional studies (Nielsen et al., 2019 and Pebody et al., 2018) and clarified the specific advantages of the FluMOMO model in detecting significant deviations from expected baseline patterns, particularly in contexts involving seasonal variations and sudden changes in healthcare system utilization. The modifications were made as follows:
“This method, which has been widely adopted in various epidemiological studies including analyses of excess mortality in Europe, [22, 23] offers a more straightforward and elegant alternative to the Farrington algorithm used in previous studies. The Farrington model, while commonly used in epidemiological studies, has limitations for our analysis due to its reliance on multiple analyst-defined parameters including seasonal adjustment factors. This introduces potential subjectivity, as different analysts may select different parameters. Furthermore, the model offers multiple methods for calculating confidence intervals, which can lead to inconsistent results. In contrast, the FluMOMO method we adopted requires no arbitrary parameter selection by analysts, ensuring more objective and reproducible results. [2, 13] It also effectively captures seasonality through sine and cosine functions with a periodicity of one and a half years.” (Materials & Methods, page 3, line 152)
Reviewer 4 Report
Comments and Suggestions for Authors
The study examines the long-term impact of the COVID-19 pandemic on healthcare utilization in Japan, addressing a highly relevant issue. The introduction clearly outlines the research objectives and context, laying the foundation for an in-depth analysis.
The methodology employed, a quasi-Poisson regression model, is suitable for examining temporal trends in healthcare utilization during the pandemic. The detailed description of data sources, variables, and statistical analyses ensures the study's replicability.
The results highlight significant trends in healthcare utilization during the pandemic. The discussion places these findings within the context of existing literature, offering possible explanations and emphasizing important policy implications for the healthcare system. The study acknowledges its limitations yet reinforces the robustness of its analyses through accurate use of citations and references.
Overall, this study provides a meaningful contribution to understanding the long-term impact of the COVID-19 pandemic on healthcare utilization in Japan. To further strengthen the work, the following areas of improvement are suggested:
- Add, as the final paragraph of the introduction, a brief explanation of how the paper is structured across its sections.
- Introduce a section 2, "Background," explaining the state of the art from recent literature. Include a subsection 2.1, "Related Studies," to mention related studies, potential gaps in the literature, and the innovative contribution of this study compared to previous research.
- In the "Materials & Methods" section, before the "Data Sources" paragraph, provide a description of the study's methodological workflow, possibly with a diagram to give an immediate visual representation of the process, accompanied by comments.
- All equations used should be numbered appropriately.
- In the "Discussion" section, add a paragraph explaining the choice of quasi-Poisson regression over machine learning predictive models. This would clarify the reasons behind the methodological preference, highlighting its advantages for the specific analysis of the study, especially in comparison to the mentioned machine learning models.
Author Response
General comments:
- The study examines the long-term impact of the COVID-19 pandemic on healthcare utilization in Japan, addressing a highly relevant issue. The introduction clearly outlines the research objectives and context, laying the foundation for an in-depth analysis. The methodology employed, a quasi-Poisson regression model, is suitable for examining temporal trends in healthcare utilization during the pandemic. The detailed description of data sources, variables, and statistical analyses ensures the study's replicability. The results highlight significant trends in healthcare utilization during the pandemic. The discussion places these findings within the context of existing literature, offering possible explanations and emphasizing important policy implications for the healthcare system. The study acknowledges its limitations yet reinforces the robustness of its analyses through accurate use of citations and references.
Thank you for your comprehensive and encouraging feedback. We greatly appreciate your positive assessment of our study's relevance, methodological approach, and analytical rigor. We have carefully addressed all your detailed comments while maintaining these strengths that you highlighted.
Major comments:
- Add, as the final paragraph of the introduction, a brief explanation of how the paper is structured across its sections.
Thank you for your kind suggestion. We have added the following brief explanation regarding the structure of our paper.
“This paper is structured to first introduce the methods and statistical approach, followed by a presentation of the results, and concluding with a discussion that includes sections on policy implications and methodological limitations.” (Introduction, page 3, line 105)
- Introduce a section 2, "Background," explaining the state of the art from recent literature. Include a subsection 2.1, "Related Studies," to mention related studies, potential gaps in the literature, and the innovative contribution of this study compared to previous research.
Thank you for the kind suggestion regarding improving the structure of the introduction. We have added a subsection for Related Studies to better introduce the literature gap addressed by our study and contextualize the greater global issues surrounding health utilization. The following text is copied below.
“Related Studies
Foregone care and essential service disruption were widely studied during the pandemic. Within the global context, it was reported by the WHO that 84% of 124 surveyed countries were still experiencing some reduction in service volume for at least one essential medical service at the end of 2022; this represented significant efforts made toward recovery, with 23% of services disrupted on average per country in late 2022 compared to 56% in 2020. [14] In the most recent global systematic review on the impact of the pandemic on healthcare utilization using data up until August 2020, a median 37% reduction in overall medical services, a 42% reduction in outpatient visits, and a 28% reduction in inpatient admissions were identified. [15] However, there is a significant gap in the literature, with few studies published on healthcare utilization recovery since late 2021 following the end of the most acute phases of the pandemic with global lockdowns, and even fewer past the declared end to the pandemic in May 2023. In one study from South Korea, a regional peer to Japan with similarly low COVID-19-associated mortality, most health services found that their utilization levels rebounded to pre-pandemic levels by late 2022. [16] On the contrary, a study from the United States (US) in a large health system in California showed the opposite pattern, with maintained exiguous utilization for inpatient and emergency services after the end of the pandemic in June 2023. [6]” (Introduction, page 2, line 68)
- In the "Materials & Methods" section, before the "Data Sources" paragraph, provide a description of the study's methodological workflow, possibly with a diagram to give an immediate visual representation of the process, accompanied by comments.
Thank you for your valuable suggestion. We have added a comprehensive methodological workflow section that provides a clear overview of our study process, from data collection through analysis. While we considered including a workflow diagram, we determined that our relatively straightforward methodology - focusing on two bed types with no specific eligibility criteria beyond the temporal and facility type parameters - could be more effectively communicated through a concise textual description that avoids redundancy with other methodology sections. The following text is copied below.
“Methodological Workflow
This study analyzed the indirect effects of the COVID-19 pandemic on healthcare utilization in Japan through the following process: First, we obtained monthly hospital report data from MHLW for general hospital and psychiatric beds from January 2012 to November 2023. The data included key healthcare delivery indicators such as inpatient volume, occupancy rate, number of hospital beds, new hospitalizations, length of hospital stays, and outpatient volume. After careful consideration of the study objectives, we excluded infectious disease beds, tuberculosis beds and long-term care beds from the analysis to focus on the pandemic's indirect effects on acute and psychiatric care utilization. Using quasi-Poisson regression modeling based on the FluMOMO approach, we then analyzed pre-pandemic data (2012-2019) to establish baseline utilization patterns and predict expected utilization levels. Finally, we compared these predictions with actual observed values during the pandemic period to quantify healthcare utilization deficits.” (Materials & Methods, page 3 line 109)
- All equations used should be numbered appropriately. (I am sorry but I dono)
Thank you very much for your comments. We added equation numbers for each equation.
- In the "Discussion" section, add a paragraph explaining the choice of quasi-Poisson regression over machine learning predictive models. This would clarify the reasons behind the methodological preference, highlighting its advantages for the specific analysis of the study, especially in comparison to the mentioned machine learning models.
Thank you for your valuable feedback. In response to this point, we have added the following explanation:
“With regards to the methodology for this study, the FluMOMO model was chosen due to the necessity to estimate expected baseline patterns of health utilization. While machine learning methods often have high accuracy in predicting average values, it is often difficult to theoretically calculate the upper and lower limits of predictions. In addition, methods using quasi-Poisson regression such as the FluMOMO model and the Farrington model have the advantage of being able to theoretically handle overdispersion appropriately and are widely used in the field of public health. (Discussion, page 11, line 331)